**Changing seasonality of moderate and extreme precipitation events in the Alps**
Stefan Brönnimann[1,2], Jan Rajczak[3], Erich M. Fischer[3], Christoph C. Raible[1,4], Marco Rohrer[1,2],
Christoph Schär[3]
[1] Oeschger Centre for Climate Change Research, University of Bern, Switzerland
[2] Institute of Geography, University of Bern, Switzerland
[3] Institute for Atmospheric and Climate Science, ETH Zurich, Switzerland
[4] Climate and Environmental Physics, Physics Institute, University of Bern, Switzerland
**Abstract.** The intensity of precipitation events is expected to increase in the future. The rate of
increase depends on the strength or rarity of the events; very strong and rare events tend to follow the
Clausius-Clapeyron relation, whereas weaker events or precipitation averages increase at a smaller
rate than expected from the Clausius-Clapeyron relation. An often overlooked aspect is seasonal
occurrence of such events, which might change in the future. To address the impact of seasonality, we
use a large ensemble of regional and global climate model simulations, comprising tens of thousands
of model years of daily temperature and precipitation for the past, present and future. In order to make
the data comparable, they are quantile-mapped to observation-based time series representative of the
Aare catchment in Switzerland. Model simulations show no increase in annual maximum 1-day
precipitation events (Rx1day) over the last 400 yrs and an increase of 10-20% until the end of the
century for a strong (RCP8.5) forcing scenario. This fits with a Clausius-Clapeyron scaling of
temperature at the event day, which increases less than annual mean temperature. An important reason
for this is a shift in seasonality. Rx1day events become less frequent in late summer and more frequent
in early summer and early fall, when it is cooler. The seasonality shift is shown to be related to
summer drying. Models with decreasing annual mean or summer mean precipitation show this
behaviour more strongly. The highest Rx1day per decade, in contrast, shows no change in seasonality
in the future. This discrepancy implies that decadal-scale extremes are thermodynamically limited;
conditions conducive to strong events still occur during the hottest time of the year on a decadal scale.
In contrast, Rx1day events are also limited by other factors. Conducive conditions are not reached
every summer in the present, and even less so in the future. Results suggest that changes in the
seasonal cycle need to be accounted for when preparing for moderately extreme precipitation events
and assessing their socio-economic impacts.

## 1.    Introduction

Heavy precipitation extremes in the Alps may trigger flood events or landslides that lead to loss of lives, cause large monetary losses and threaten important infrastructure. While the most extreme events have the highest socio-economic impact, even moderate extreme events such as the annual maximum rainfall (Rx1day) or events with a 10-yr return period may be relevant for climate change adaptation. Even for these events, the observation records at a single location may not be long enough to capture trends. Still, aggregating weather station records from Switzerland, Scherrer et al. (2016) found an increase in intensity as measured by the magnitude of the annul maximum 1-day precipitation extreme (Rx1day) or exceedances of the 99[th] percentile.

Heavy precipitation requires moisture convergence and convection or synoptic-scale uplift. With increasing temperatures, saturation specific humidity increases and as a consequence an increased precipitation intensity is expected. At the global scale, precipitation is limited by radiation that has to balance the latent heat release (e.g. Allen and Ingram 2002). For extreme precipitation events, however, Scherrer et al. (2016) found that the increase closely follows a Clausius-Clapeyron scaling of the annual mean temperature trend. Fischer and Knutti (2016) also found a close to Clausius-Clapeyron scaling to regional temperature changes both in global models and observations. However, the scaling may not hold exactly for various reasons (Pendergrass 2018).

Firstly, global models do not resolve changes in convection (Prein et al., 2015; Zhang et al., 2017), which is important for the case of Alpine precipitation (Giorgi et al., 2016). For instance, trends in Alpine precipitation are different during summer in convection-resolving regional models (Ban et al., 2014). Secondly, the scaling of extreme precipitation with temperatures at day-to-day time scales cannot generally be extrapolated to the future based on annual or seasonal mean temperatures (e.g., Ban et al. 2015; Schär et al 2016; Zhang et al. 2017). One possible cause for a discrepancy between scaling at day-to-day time scales and at time scales of long-term warming is a potential change in seasonality. Pfahl et al. (2017) in an analysis of CMIP5 data (Taylor et al., 2012), found a shift in the future Rx1day towards smaller saturation specific humidity over most of the northern extratropical land areas (their Fig. S6), indicative of lower temperatures and thus a shift towards the colder seasons. Recently, Messmer et al. (submitted) focused on one dynamical feature, which is responsible for extreme precipitation events in the Alps, the so-called Vb cyclone (van Bebber, 1898; Messmer et al., 2015). They found in dynamically downscaled scenario simulations (RCP 8.5) that the occurrence of extreme Vb cyclones is shifted from the midsummer to May and September in the future.

In this paper we focus on seasonality changes of heavy precipitation events (Rx1day) in the Alps using a comprehensive set of global and regional model simulations comprising tens of thousands of years of daily temperature and precipitation quantile mapped to observations representative of a large catchment in Switzerland. Based on this comprehensive data set, we analyse (1) trends in Rx1day and

their relation to temperature trends, (2) the effect of changes in the seasonal cycle of temperature on
Rx1day events and (3) the effect of changes in the seasonal frequency of occurrence of Rx1day.
The paper is organised as follows. Section 2 gives a brief overview of the data and the quantile
mapping approach (Rajczak et al., 2018) and outlines the analyses performed. Section 3 presents the
results for Rx1day. A discussion in terms of underlying mechanisms and differences between models
and model set-ups follows in Section 4. Conclusions are drawn in Section 5.

## 74  2.    Data and Methods

The study focuses on the Aare river catchment in Switzerland, an area of approximately 17 000 km$^2$
(see Fig. 1). Heavy precipitation events in this catchment can cause major floods of the lower Aare
river and the Rhine, where several nuclear power plants are located. We extract daily temperature and
precipitation data over this domain from a large data set, comprising simulations of the past, present
and the future from different set-ups (coupled and uncoupled simulations, global and regional
simulations, single member or ensemble simulations), reanalyses, dynamically downscaled reanalyses,
and observations. A total of 55,000 simulation years are available (Rajczak et al., 2018, the data are
available from this website: https://doi.pangaea.de/10.1594/PANGAEA.886881). In this study we
focus on experiments with regional and global models. The following ensembles are used:
• CCC400: An ensemble of 30 AMIP-type global GCM simulations using ECHAM5.4 at T63
and covering the period 1600–2005
• the CMIP5 ensemble of 25 simulations (historical and RCP8.5) covering the period 1900–
87     2100
• initial condition ensemble of 21 COSMO simulations (0.44°) using the RCP8.5 scenario and
covering the years 1950–2100
• a set of 13 ENSEMBLES simulations using the A1B scenario and covering the period 1971–
91     2100
• a set of 15 CORDEX-11 simulations at 0.11° resolution (RCP8.5 and RCP4.5) covering the
period 1970–2099
• a set of 17 (RCP4.5) and 19 (RCP8.5) CORDEX-44 simulations at 0.44° resolution covering
the period 1970–2099
Table 1 gives an overview of all experiments used in our study, including references. The fact that the
data sets differ in several key aspects (e.g., resolution, time period covered, scenario used, complexity
of the model) allows a comprehensive view of possible changes and sources of uncertainties.
A consistent analysis of the available data sources is hardly feasible, as the data stems from models
with vastly different resolution and characteristics. All models suffer to some extent from biases. The
variety of modelling approaches thereby implies significant model-to-model differences. For this
reason, we focus on a fairly large catchment (where all models should arguably have at least some
potential), and use a statistical approach (quantile mapping) to calibrate the simulation results against
observations.
More specifically, the data are quantile-mapped (Themessl et al., 2011; Gudmundsson et al., 2012;
Teutschbein and Seibert, 2012; Räty et al., 2014) to observation-based time series representative of the
average over the Aare catchment in Switzerland (see Fig. 1 for the stations; Fig. S1 shows the
observation-based, annual time series). Both data sets are used with daily resolution. The data and bias
correction are described in detail in Rajczak et al. (2018). The method has recently been used in other
Swiss climate impact studies (Rajczak et al., 2016a), and is reasonably skilful in daily and multi-day
precipitation diagnostics (Rajczak et al., 2016b). The transfer-function depends on the season and is
based on a 91-day moving window, centred over the day of the year (Themessl et al., 2012; Wilcke et
al., 2013; Rajczak et al., 2016a, 2016b). Values above the observed range of values are corrected
according to the 99.9$^{th}$ percentile (p99.9) in a constant manner (p1 and p99 in the case of temperature).
Studies recommend such an implementation opposed to complex extrapolation methods (Gutjahr and
Heinemann, 2013; Ivanov and Kotlarski, 2017; Themessl et al., 2012). Comparisons of raw and
quantile-mapped data are shown in Fig. S2. Note that quantile mapping does affect mean temperature
as well as temperature at the event day, which implies changes in saturation specific humidity. It is
therefore important to perform all analyses also for the raw data.
The analysis focuses on Rx1day, the maximum Rx1day per decade, annual mean temperature,
temperature at the event day, and the day of year of the event. All analyses are applied to each
individual simulation; only then the ensemble statistics are formed. No further weighting is performed
for multi-model data sets, but for some analyses the simulations are separated into those exhibiting a
drying or a wetting. We have performed all analyses for both the raw and quantile-mapped data. We
refer to the raw data occasionally in the text, more comprehensive material is added to the
supplementary material. For our analyses the following approach is used:
(i) We analyse long-term changes in Rx1day in each data set and compare this to the long-term
changes in annual mean temperature and as well as the trend in temperature at the event day. This
gives an indication of the proximity of trends in precipitation changes to a Clausius-Clapeyron scaling.
(ii) Changes in the seasonal cycle of temperature and event frequency (see below) are then addressed
by partitioning the data sets into sub-periods. For simulations of past climate (CCC400), we partition
the data from 1601–2000 into century chunks. For the present and future, for reasons of consistency,
we show results only for the two 35-yr periods 1971–2005 and 2065–2099, which are common to all
model experiments. In the supplementary material we also show results for all future simulations
partitioned into three periods of equal length (four periods for the longer simulations CMIP5 and
CCC400) to make full use of all data.
(iii) Within the periods, we analyse the seasonal cycle of the relative frequency of Rx1day events as
well as the seasonal cycle of temperature at the event day. The former indicates whether the
seasonality of Rx1day events changes, which is likely to affect the temperature trend on event days.
The latter takes this seasonality shift into account and indicates whether for a given calendar day,
temperature trends on event days differ from trends on all days. The annual cycle of temperature for
all days and for event days is estimated by fitting the first two harmonics of the seasonal cycle.
(iv) Finally, we address the dependence of Rx1day seasonality changes on background climate trends.
This is done for multi-model ensembles by stratifying the simulations within an ensemble according to
their linear trend (obtained with least-squares regression) in annual or summer mean precipitation over
the period 1971–2099. This allows addressing common signatures, *e.g.,* whether drying models tend to
show stronger changes in seasonality than wetting models.

**3.    Results**
Before analyzing the statistics of Rx1day, a short description of typical Rx1day events at the
catchment scale are provided. Figure S2 shows meteorological fields from the CERA20C reanalysis
(Laloyaux et al. 2018) for the three strongest Rx1day events. The ten strongest events are listed in
Table 2, along with references. The typical catchment-scale Rx1day event is caused by the passage of
a cold front related to an elongated trough or cut-off low situation with destabilisation and pre-frontal
convective activity. Such situations are often associated with convergence and lifting of moist air,
originating from the Mediterranean region, north of the Alps in a 'Vb'-type flow situation (see also
Stucki et al. 2012, Messmer et al. 2015, 2017). The event in 1978 is a typical example (Courvoisier,
1978; Stucki et al., 2017), but the situations in 1954 and 2007 as well as many of the events listed in
Table 2 were similar (Schmutz et al., 2008). Although models may not well reproduce the flow
deformation by the Alps as well as orographic enhancement of convection, they are assumed to
reproduce the synoptic scale processes such as frontal systems, moisture transport, and uplift.
Time series of Rx1day for the ensembles CCC400 (1600–2005), CMIP5 (historical and RCP8.5,
1900–2100), the COSMO initial condition ensemble (RCP8.5, 1950–2100), the ENSEMBLES
simulations (A1B, 1971–2100), CORDEX-11 (RCP8.5, 1970–2099) and CORDEX-44 (RCP8.5,
1970–2099) are shown in Fig. 2 (top). Over the past 400 years, no change in Rx1day is found (the
same is true for past millennium simulations (Lehner et al., 2015) or the Twentieth Century Reanalysis
20CRv2c (Compo et al., 2011) included in Rajczak et al. (submitted); not shown). A slight increase in
Rx1day over the past 50 years appears in CMIP5 (historical). Note, that there are also indications at
the continental to global scale that the majority of RCMs and GCMs tend to underestimate the
observed intensification in heavy rainfall (Fischer and Knutti, 2016; Borodina et al., 2017).
Simulations for the future show a clear increase in Rx1day of around 10-20% in the ensemble mean.
In the CMIP5 ensemble, the upper range of the ensemble shows a stronger increase than the mean.
Annual mean temperature (Fig.2, bottom, red line) shows a pronounced increase of 4-6 °C in the 21st
century. According to the Clausius-Clapeyron relation, this would correspond to a 30-50% increase in
precipitation extremes, which is not the case. However, if we consider only the temperature on the day
of the Rx1day precipitation events (Fig. 2, bottom, full blue line, note that this time series is
smoothed), we find a smaller temperature increase for most of the simulations. For the 21st century, the
temperature increase during Rx1day events amounts to only 3 °C (CMIP5) or even just 1 °C
(COMSO), but with considerable differences between experiments. Trends in Rx1day of 10-20% thus
approximately follow the Clausius-Clapeyron scaling. The same analyses for raw data gives very
similar results (shown in Fig. S2).
As already stated above, the Clausius-Clapeyron argument is expected to apply for the largest events,
but not for intermediate events, and may be not even for Rx1day events. Ban et al. (2015) analyzed
convection-resolving climate simulations at a resolution of 2 km, and calculated the scaling rates as a
function of all-day percentiles in the Alpine region for summer (see their Fig. 4e). In terms of
percentiles, our Rx1day event class would correspond to roughly the 99.7th percentile, and at this event
category the scaling estimate of Ban et al. (2015) amounts to about 1-3%/°C. This is roughly
consistent with the results in Fig.2 (when using mean temperature changes). Nevertheless, Ban et al.
(2015) found that increases in precipitation intensity would approximately asymptote towards the
Clausius-Clapeyron rate at very high percentiles.
The lower part of Fig. 2 also shows the same analysis for the temperature taken at the highest Rx1day
per decade (dashed lines). Note that sample size here is very small except for CCC400 and CMIP.
Interestingly, in CMIP5 temperatures during these events increase more strongly than for all Rx1day
events. The trend resembles that of the annual mean.
Two factors can contribute to the fact that the temperature increase on Rx1day events is smaller than
the increase in the annual mean: (1) a change in the occurrence frequency of Rx1day events towards a
colder season or (2) a different change of the temperature on Rx1day events even for unchanged
seasonality. We first analyse the latter. Considering the seasonal cycle of temperature during Rx1day
events (Fig. 3, top row), we find that in summer, Rx1day events in the present occur on days that are
slightly colder than average, while those few Rx1day events that occur during the cold season occur
with warmer than average conditions. This is evident in both observations (long black dashes) and
model simulations. In the future, Rx1day events tend to occur on days that are even cooler than
average, *i.e.,* the trend on Rx1day events (the difference between 2065–2099 and 1971–2005 is shown
on the right) is ca. 1 °C smaller than the trend on all days. This holds for all model ensembles, and the

difference has no obvious seasonal cycle. Thus, part of the trend difference between Rx1day and all days is unrelated to the seasonality of Rx1day.

The day-of-occurrence of Rx1day (Fig. 3, second row) shows a broad summer peak in the observations. Several of the model simulations simulate two peaks, one in early summer (June) and one in early fall (September). Both in the past (CCC400) and in the future, there is a trend towards fewer occurrences during June and July and more during the neighbouring months, *i.e.,* both the early summer and the early fall peaks shift towards the cold season. This becomes particularly clear when plotting the difference in relative frequency as a function of day of year between 2065–2099 and 1971–2005 (Fig. 3, middle right) for each experiments. The mid-summer events become rare; their occurrence already decreased by 10% during the past 400 years and might further decrease by 30% in the future. This shift in seasonality explains the remaining difference in temperature trends between Rx1day event days and all days.

As we use the periods 1971–2005 and 2065–2099, Figure 3 (second row) is based on 100 (left) or 35 (middle) values per ensemble member and results were then smoothed with a Gaussian kernel with a bandwidth of 15 days. In order to exploit the full data set, we also partitioned each data set into equal-sized bins (Fig. S3) and found very similar results. Note, also, that the change in seasonality is not due to the quantile mapping but appears also in the raw data (see Fig. S2).

However, the above results depend on the rarity of events. When analyzing rarer events, such as the highest Rx1day events per decade (Fig. 3, bottom; note that samples are large enough only for CCC400 and CMIP5, results for other experiments are shown in Fig. S3), we find a different result. These events are even more concentrated to mid-summer than Rx1day in the present climate (the mid-summer dip almost vanishes). In CCC400, their occurrence is high throughout the summer whereas in CMIP5 these are mostly early fall events. Interestingly, there is no change in the frequency of occurrence over time. Hence, only moderately extreme events (with a frequency of once per year) are affected by a change in seasonality and not the more extreme events.

What causes this change in seasonality? Given the limited number of variables at our disposition (temperature, precipitation), it is not possible to properly disentangle dynamical and thermodynamical contributions. However, analyzing differences between the ensemble members gives important indications. Within ENSEMBLES, CORDEX-11, CORDEX-44, and CMIP5 the contributing ensemble members are grouped according to their trend in annual or summer mean precipitation (Fig. 4) and the analysis of the frequency of occurrence is repeated. Clear differences are found between ensemble simulations, which show a drying and ensemble members with a positive trend in precipitation in summer and in the annual mean (Fig. 4), particularly in CMIP5 and CORDEX-44, less so in ENSEMBLES and CORDEX-11. The wetting ensemble members show hardly any change in seasonality or even an increased frequency in summer, while the drying members show a decrease in

the frequency of occurrence of Rx1day events in mid-summer. The inter-model variability is large as
the samples get smaller, but the signature is robust across the entire data set (see Fig. S4).
Drying might also explain the changing seasonality in CCC400, as the ensemble mean shows
decreasing precipitation over the past 400 years (Fig. S6). Note that a mis-specified trend in land-
surface parameters might affect CCC400 (Rohrer et al., 2018). However, a simulation with corrected
land surface shows the same change in seasonality (Fig. S3).

**4.     Discussion**
Heavy precipitation events (Rx1day) in the Aare catchment in the present climate are most frequent
during the warm season. If all preconditions (atmospheric circulation, stability, soil moisture, etc.) are
conducive to a heavy precipitation event, the intensity of the event is then expected to depend mainly
on thermodynamics, *i.e.,* on saturation specific humidity. It will thus follow a Clausius-Clapeyron
scaling as a first order approximation, often summarized as thermodynamic changes. In fact, trends in
Rx1day over the Aare catchment do follow a Clausius-Clapeyron scaling, however not of annual mean
temperature but rather of event-day temperature. This is because Rx1day may also change due to
dynamic changes e.g., conducive conditions become less frequent and are not reached each summer.
Such changes may relate to changes in the large-scale circulation patterns or enhanced atmospheric
stability (e.g. Kröner et al., 2017) leading to reduced vertical updrafts (Pfahl et al., 2017).
In this respect, Rx1day and decadal-scale events exhibit a different behaviour in that the former show
a change in seasonality while the later do not. This discrepancy may be interpreted in the following
way: Decadal-scale precipitation extremes are mainly thermodynamically limited, *i.e.,* chances are
high that within a decade, conducive conditions occur on a late summer day, when temperatures and
thus saturation specific humidity are highest. In the future, such conditions can still be reached in late
summer once per decade, and thus the increase in temperature leads to an intensification of the most
extreme events, with no apparent change in seasonality, consistent with Ban et al. (2015). However, on
an annual scale such conditions are not reached every summer. Even in the present climate, Rx1day
events do not occur most frequently at the warmest time of the year. This indicates that other factors
(which we termed preconditions above) matter and Rx1day events are not fully thermodynamically
limited. This tendency will strengthen in the future as preconditions change, leading to a change in
seasonality. This result is consistent though more pronounced than identified by Pfahl et al. (2017)
who found a change of Rx1day events towards lower saturation specific humidity over the northern
extratropical land areas, indicating that the behaviour found in our analysis may be true for other
extratropical land regions as well.
While we cannot disentangle the contributions of individual factors to the preconditions, we can
identify drying as one if the important factors in models. Models that show a drying trend tend to
exhibit a shift in the seasonal cycle of moderate extremes (Rx1day) away from late-summer towards
early summer and fall, whereas the others do not.
A further result of our study is that even for a given calendar day, the future temperature trend in the
models is smaller for Rx1day than for all days. A possible explanation, particularly for summer days,
is model drying and thus excessive heating on non-precipitation days. In fact, stratifying the
simulations according to their summer-mean precipitation trends indicates that drying simulations
have a stronger cooling of event days relative to all days (Fig. S5).
The findings of our study - intensification of the most extreme events and change of seasonality for
moderate extremes - is relevant for adapting to future climatic changes. This is particularly the case as
observational coverage of the highest Rx1day per decade is limited. Whether or not the seasonality
shift is a real effect or occurs due to artificial summer drying of the models remains to be studied.
More detailed studies of the underlying processes, including stability and atmospheric circulation,
during future extreme events are also required (*e.g.,* Messmer et al. submitted).

**5.    Conclusions**
Over the Alpine region moderate precipitation extremes such as characterized by Rx1day may not
increase as much as expected from applying a Clausius-Clapeyron to the change in annual mean
temperature, due to a change in seasonality of Rx1day events, and due to smaller-than-average
temperature trends during event days (regardless of the season). Both reasons are due to the fact that
moderate extreme events are not purely thermodynamically limited. In our study, we find that Rx1day
events in a future climate tend to occur less frequently in mid-summer, but more often in spring and
autumn. A similar change is also found over the past 400 years in model simulations. Further analyses
show that mostly those models are concerned whose annual mean or summer mean precipitation
decreases, i.e., the changing seasonality is in part due to drying. Conversely, 10-yr events do exhibit
their highest frequency in mid-summer also in the future, with no apparent change in seasonality.
For flood protection this means that moderate events might shift towards the cold season with only a
small change in intensity, but the more relevant extreme events such as those with 10-yr return period
remain in summer and increase strongly in intensity.

**Acknowledgements.** This work was funded by the Swiss Federal Office for the Environment (FOEN),
the Swiss Federal Office of Energy (SFOE), and the Swiss Federal Nuclear Safety Inspectorate (ENSI)
in the framework the project EXAR: Understanding Extreme Flooding Events Aare-Rhein in
Switzerland as well as by the Swiss National Science Foundation (project 200021_143219 "EXTRA-
LARGE"). We acknowledge the World Climate Research Programme's Working Group on Regional

Climate, and the Working Group on Coupled Modelling, former coordinating body of CORDEX and responsible panel for CMIP5. We also thank the climate modelling groups (listed in Table S1 of this paper) for producing and making available their model output. We also acknowledge the Earth System Grid Federation infrastructure an international effort led by the U.S. Department of Energy's Program for Climate Model Diagnosis and Intercomparison, the European Network for Earth System Modelling and other partners in the Global Organisation for Earth System Science Portals (GO-ESSP). Compute facilities for the CCC400 simulations were provided by the Swiss National Supercomputing Centre (CSCS). The Twentieth Century Reanalysis Project is supported by the U.S. Department of Energy (DOE) Office of Science Innovative and Novel Computational Impact on Theory and Experiment (INCITE) program, and Office of Biological and Environmental Research (BER), and by the National Oceanic and Atmospheric Administration Climate Program Office

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

**Tables**

**Table 1.** Overview of model experiments used in our study (see Table S1 for further details)

| Experiment | Type | Resol. | $n$ | Period | Scenario | Reference |
|---|---|---|---|---|---|---|
| CCC400 | Global GCM | 2° | 30+1* | 1600–2005 | Historical | Bhend et al. (2012) Rohrer et al. (2018) |
| CMIP5 | Global AOGCM | various | 25 | 1901–2100 | Historical, RCP8.5 | Taylor et al. (2012) |
| ENSEMBLES | Regional | 25 km | 13 | 1970–2099 | A1B | van der Linden and Mitchell (2009) |
| COSMO | Regional | 0.44° | 21 | 1950–2100 | RCP8.5 | |
| CORDEX-11 | Regional | 0.11° | 15 | 1971–2099 | RCP8.5 | |
| CORDEX-44 | Regional | 0.44° | 19 | 1971–2099 | RCP8.5 | Jacob et al. (2014); Kotlarski et al. (2014) |
| CORDEX-11 | Regional | 0.11° | 15 | 1971–2099 | RCP4.5 | |
| CORDEX-44 | Regional | 0.44° | 17 | 1971–2099 | RCP4.5 | |

\* Trends in some land-surface properties were mis-specified in the 30-member ensemble, an additional member with corrected land-surface properties was performed to assess the errors (Rohrer et al., 2018). The figures in this paper show the 30 member ensemble; results for the land-surface corrected member share the same features and are shown in Fig. S3.

**Table 2.** The largest ten Rx1day events in the catchment-averaged observations sorted by their strength.

| Year | Mon | Day | Rx1day (mm) | Comment |
|---|---|---|---|---|
| 1978 | 8 | 7 | 84.8 | 62% of Swiss area >70 mm/d (Courvoisier et al. 1979), flooding of Rhine (Stucki et al. 2017), max Rx1day at several stations in the region (MeteoSwiss, 2006) |
| 1954 | 8 | 21 | 76.5 | 39% of Swiss area >70 mm/d (Courvoisier et al. 1979) |
| 2007 | 8 | 8 | 64.0 | large flooding (Bezzola and Ruf, 2009), highest runoff at Aare (Untersiggental) |
| 1908 | 5 | 23 | 55.0 | max Rx1day at several stations in the region (MeteoSwiss, 2006) |
| 1939 | 8 | 5 | 54.9 | max Rx1day at one station in the region (MeteoSwiss, 2006) |
| 2009 | 7 | 17 | 54.2 | large damages due to (prefronal) thunderstorms |
| 1977 | 7 | 31 | 53.9 | max Rx1day at one station in the region (MeteoSwiss, 2006) |
| 1991 | 12 | 21 | 53.5 | max Rx1day at several stations in the region (MeteoSwiss, 2006) |
| 2005 | 8 | 21 | 53.1 | major flood event in Switzerland (MeteoSwiss, 2006) |
| 1910 | 6 | 14 | 52.4 | major flood event in Switzerland (Stucki et al. 2012), 25% of Swiss area >70 mm/d (Courvoisier et al. 1979), max Rx1day at several stations in the region (MeteoSwiss, 2006) |

**Figures**

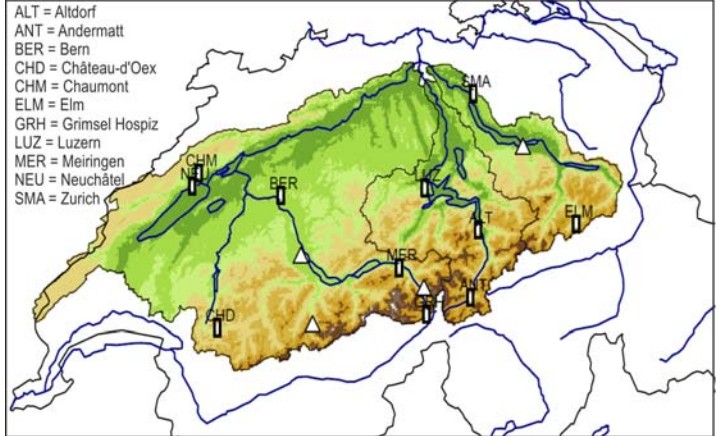


**Fig. 1**: Catchment of the Aare river. All data used in this study represent averages over this region.
The representative weather stations are shown as squares (providing both temperature and
precipitation) and triangles (only precipitation).

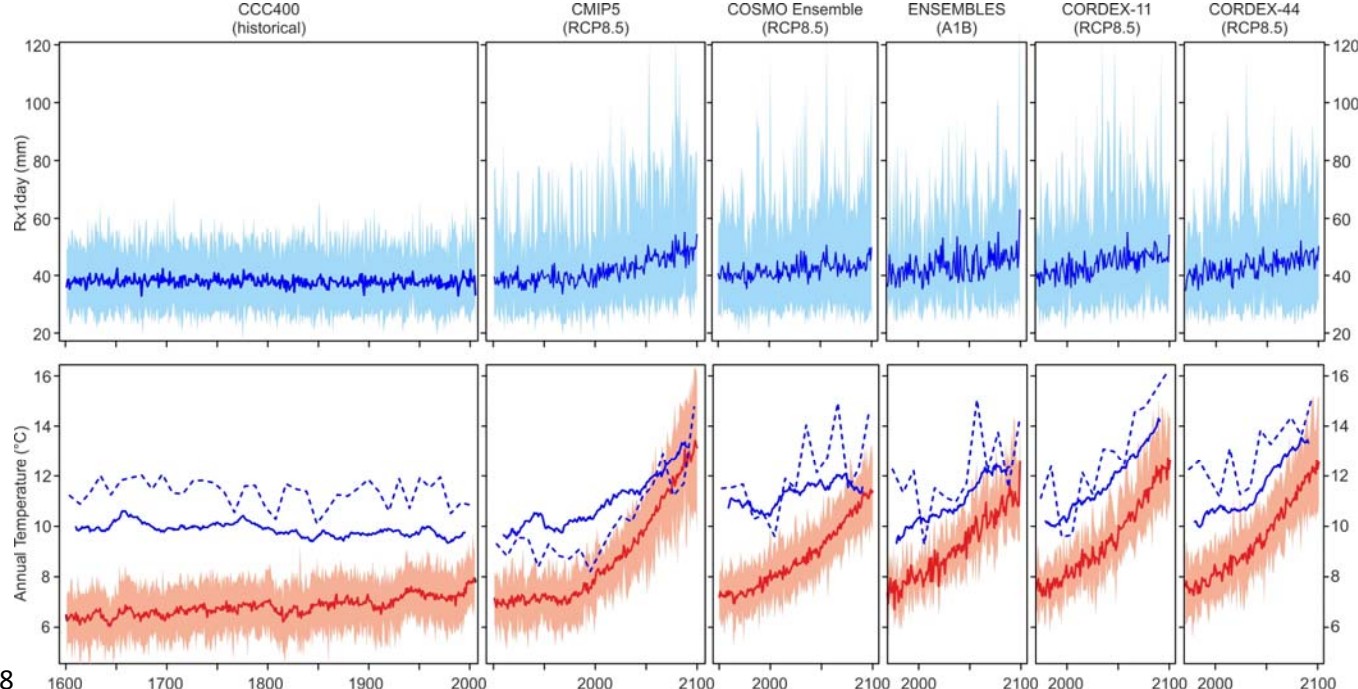


**Fig. 2:** Annual time series (ensemble mean as line and ensemble range as shading) of (top) Rx1day
and (bottom) annual mean temperature (red) as well as the temperature during the Rx1day event (blue
solid line, smoothed with a 20-yr running average) in CCC400, CMIP5 and COSMO (quantile
mapped). Also shown is the temperature at the highest Rx1day event per decade (blue dashed;
ensemble mean).

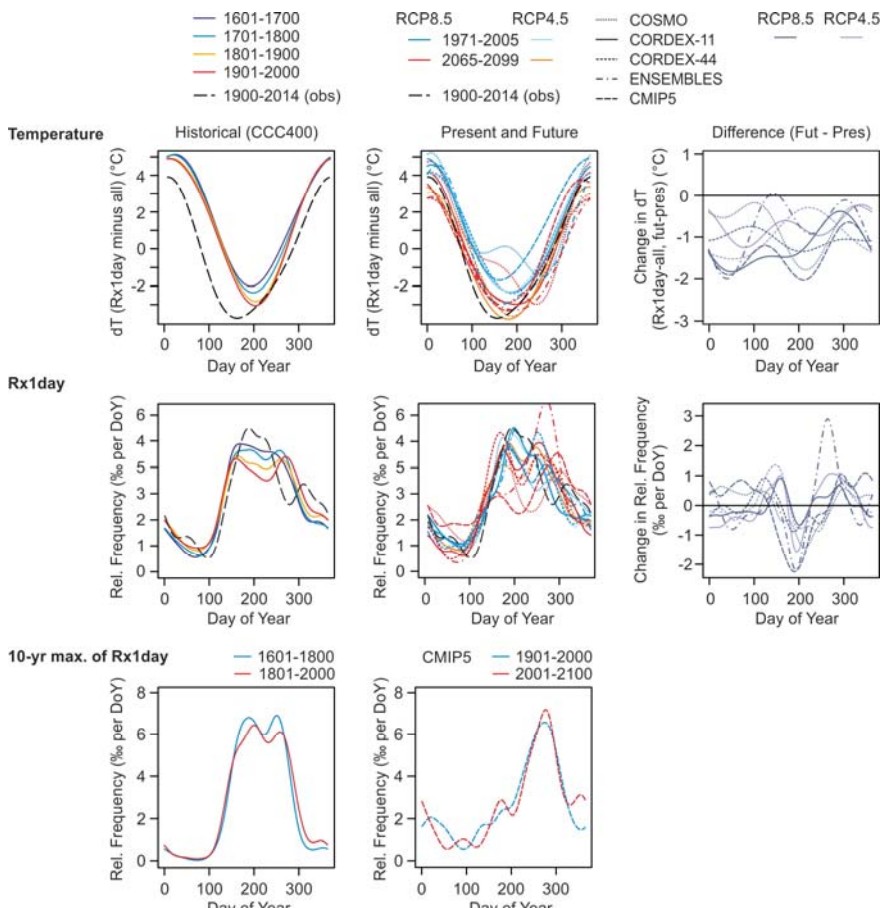


**Fig. 3:** (top row) Temperature difference as a function of calendar day between event days (Rx1day) and all days for different time periods in CCC400 (left) and in model simulations of the present (1971–2005) and future (2065–2099) (middle; the right figure shows the difference between the two periods). (middle row) Density plot of the day of occurrence of Rx1day events for different time periods in CCC400 (left) and in model simulations of the present (1971–2005) and future (2065–2099) (middle; the right figure shows the difference between the two periods). (bottom row) Same as second row, but only for the highest Rx1day per decade. Note that only CCC400 and CMIP5 have sufficiently large ensembles, and longer time periods were chosen. All analyses are based on quantile-mapped data (see Fig. S3 for additional plots and Fig. S2 for the same analysis based on raw data). A Gaussian Kernel smoother with a bandwidth of 15 days was used for plotting. Results from catchment-averaged observations are shown as long black dashes.

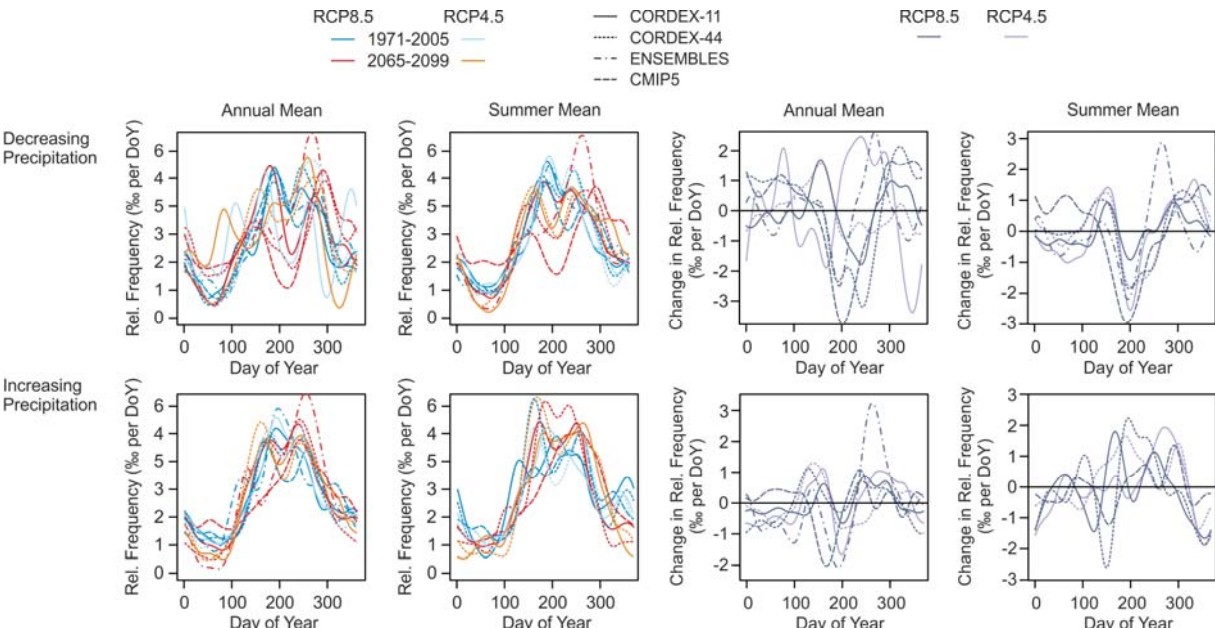

**Fig. 4:** Density plot of the day of occurrence of Rx1day for the present (1971–2005) and future (2065–2099) in different the multi-model ensembles (left two columns). The right two columns show the difference between future and present climate. Ensemble members are separated into those that show a positive or negative trend in annual mean (first and third column) or summer mean (second and fourth column) precipitation over the 1971–2099 period (see Fig. S4 for additional plots from other scenarios and different time periods).