# Peer review of "Changing seasonality of moderate and extreme precipitation events in the Alps"

_Natural Hazards and Earth System Sciences, 2018_

## Referee Comment (RC1) · Anonymous Referee #1 · 28 Mar 2018

General:

Brönnimann et al. analyse the seasonal cycle and temperature dependence of the annual (Rx1day) and decadal precipitation maximum for a catchment in the Swiss Alps under present day and climate change conditions. They evaluate a large ensemble of global and regional model simulations comprising 555,000 simulation years. The more moderate Rx1day events exhibit a change in seasonality from mid-summer towards spring and fall under future climate conditions, especially in models which simulate increasingly dry conditions for the summer season. Meanwhile, decadal maxima remain most frequent during summer and increase in strength. The authors conclude that only the more extreme precipitation events (with return periods of 10-years or higher) are thermodynamically limited. These are also the events with exhibit approximate Clausius-Clapeyron scaling. More moderate extremes may be affected by less frequent conducive preconditions.

The topic is of relevance to the scientific community and within the scope of NHESS. The paper is well written and easy to follow. The conclusions are sound. I recommend publication after some minor, mostly technical revisions.

Specific comments:

It would be nice to include a figure on the seasonal cycle of Rx1day in the observations in the supplement with a comment in the manuscript.

Technical remarks:

Line 12 : Please state if weaker events show under or over the Clausius-Clapeyron scaling.

Line 107: "bias-correction" instead of "downscaling"

Line 107: The paper of Rajczak et al. 2018 is not accessible

Line 234: one "on" too many

Line 417: ...annual mean temperature (red) ...

―――――――――――――――

---

## Referee Comment (RC2) · Anonymous Referee #2 · 6 Apr 2018

Brönnimann et al. present an impressive and sound study on expected changes in extreme precipitation in the Aare catchment. Their major results are an expected future increase in rx1day (but less increase than might be expected by thermodynamic constraints), a change in seasonality for moderate extremes and no change in seasonality for rarer extremes. In addition, the authors present an interesting discussion and interpretation of their results.

There are two points that let this study stick out from many other papers: Firstly, the authors analyzed a huge ensemble of very different (global and regional) climate models of different generations. This enables them to draw much more robust conclusions than most other studies, since any common feature of such a divers set of simulations is only very unlikely due to particular model deficiencies and more likely due to physical

changes in the system. Secondly, the authors analyse the raw model output along with the bias corrected one. Since their main results are deducible from both versions of the dataset, this again increases credibility. As a side effect, they deliver very interesting material for the ongoing discussion about the limits of applicability of bias correction in climate change studies.

Thus I clearly suggest this study for publication after some minor changes/additions as follows:

1) The study analyses precipitation and temperature averages over a fairly large area with diverse geographical features. They clearly explain why they do so and I'm fine with it. However, when looking at 1-day precipitation extremes, there are likely to be quite some differences between those at station scale (that the reader might intuitively think of, when reading this study) and those averaged over 17000 square kms. I'm quite sure that we are not talking about the local convective systems that move slowly and therefore often bring extreme precipitation amounts locally, while neighboring stations are not affected. What kind of meteorological situations are we talking about? Probably frontal systems that move over the entire region? Could you please discuss this, to give the study the right framing? E.g. a typical example of an area-wide rx1day event opposed to a typical station-scale rx1day event would be very instructive. This is not mandatory, but at least a few sentences on the differences between station scale and area-average should be added.

2) One of your major interpretations of the results is, that thermodynamic constraints are not the dominating constraints for moderate extremes, but for rarer extremes (10 year's rx1day) thermodynamic constraints dominate. This would mean: The hotter, the more rain, right? If this is the case, why has the annual cycle of rare extremes a notch during the hottest phase of the year? (Figure 3, bottom left panel). Isn't that a contradiction? Please comment on that.

3) Editorial:

[Figure]

Line 66: "(2) changes in the seasonal cycle of temperature on Rx1day events". Something is missing here. Maybe "the effect of" in the beginning?

Line 81: "In this study we focus on experiments with regional or global models." regional AND global models?

Line 141: "(iv) Finally. . ." This sentence hard to comprehend. After looking at the results, it becomes clear what you mean, but please consider rephrasing this sentence for better comprehensibility.

Fig S2: In the figure caption, there is a "Top:" to much.
* * *

---

## Author Comment (AC1) · 7 Jun 2018

Specific comments

"It would be nice to include a figure on the seasonal cycle of Rx1day in the observations in the supplement with a comment in the manuscript."

This is a very good comment. In the revised manuscript we will add the seasonal cycle of the observations (not only Rx1day, but also the temperature difference between event day and all days). But wqe will do that in the main manuscript, as additional lines to Figure 2. This shows that CCC400 is doing less well in reproducing the seasonal cycle than some of the model simulations (ENSEMBLES, CORDEX-11). We will also add text to the corresponding places in the manuscript.

[Figure]

Technical remarks:

"Line 12 : Please state if weaker events show under or over the Clausius-Clapeyron scaling."

The scaling is smaller – this sentence will be changed in the revised manuscript.

"Line 107: 'bias-correction' instead of 'downscaling'"

Changed.

"Line 107: The paper of Rajczak et al. 2018 is not accessible"

The reference will be changed; the submitted manuscript will be uploaded with the revised version of this manuscript.

"Line 234: one 'on' too many"

Thanks.

"Line 417: ...annual mean temperature (red) ..."

Thanks.
* * *

---

## Author Comment (AC2) · 7 Jun 2018

"1) The study analyses precipitation and temperature averages over a fairly large area with diverse geographical features. They clearly explain why they do so and I'm fine with it. However, when looking at 1-day precipitation extremes, there are likely to be quite some differences between those at station scale (that the reader might intuitively think of, when reading this study) and those averaged over 17000 square kms. I'm quite sure that we are not talking about the local convective systems that move slowly and therefore often bring extreme precipitation amounts locally, while neighboring stations are not affected. What kind of meteorological situations are we talking about? Probably frontal systems that move over the entire region? Could you please discuss this, to give the study the right framing? E.g. a typical example of an area-wide rx1day event

opposed to a typical station-scale rx1day event would be very instructive. This is not mandatory, but at least a few sentences on the differences between station scale and area-average should be added."

The reviewer is right that at single stations, Rx1day may occur during highly convective situations. For Rx1day over the catchment studied, a combination of a frontal systems with prefrontal convective precipitation is often responsible. We will add some text to the paper and will add a figure to the supplementary material showing meteorological fields for the three strongest events in the observations. In the main text we will give references to the most extreme events studied and will add a table summarizing the ten largest Rx1day in the catchment. Thanks for the comment.

"2) One of your major interpretations of the results is, that thermodynamic constraints are not the dominating constraints for moderate extremes, but for rarer extremes (10 year's rx1day) thermodynamic constraints dominate. This would mean: The hotter, the more rain, right? If this is the case, why has the annual cycle of rare extremes a notch during the hottest phase of the year? (Figure 3, bottom left panel). Isn't that a contradiction? Please comment on that."

The reviewer is right that a notch is not expected. What we see here is just a tiny notch, much smaller than for Rx1day, and this is the main point here. We will add a comment on that to the revised manuscript.

"3) Editorial:

Line 66: '(2) changes in the seasonal cycle of temperature on Rx1day events'. Something is missing here. Maybe 'the effect of' in the beginning?"

We will add 'the effect of'.

"Line 81: 'In this study we focus on experiments with regional or global models.' regional AND global models?"

This will be changed.

"Line 141: '(iv) Finally...' This sentence hard to comprehend. After looking at the results, it becomes clear what you mean, but please consider rephrasing this sentence for better comprehensibility."

The sentence will be rephrased.

"Fig S2: In the figure caption, there is a 'Top:' too much."

This will be changed.

———————————————————